# Improving Data Security with Blockchain and Internet of Things in the Gourmet Cocoa Bean Fermentation Process [note 1]

**DOI:** 10.3390/s22083029

**Published:** 2022-04-15

**Authors:** Jauberth Abijaude, Péricles Sobreira, Levy Santiago, Fabíola Greve

**Affiliations:** 1Computer Science Institute, Federal University of Bahia, Salvador 40170-110, Bahia, Brazil; levy.santiago@ufba.br (L.S.); fabiola@ufba.br (F.G.); 2Exact Science Departament, Santa Cruz State University, Ilheus 45662-900, Bahia, Brazil; 3Computer Science and Engineering Department, University of Quebec at Outaouais, Gatineau, QC J8Y3G5, Canada; pericles.delimasobreira@uqo.ca

**Keywords:** smart agriculture, Internet of Things, blockchain, smart contracts, DApp, middleware, gourmet cocoa, remote monitoring, web systems

## Abstract

Brazil was one of the largest cocoa producers in the world, mainly supported by the South of Bahia region. After the 1980s, the witch-broom disease demolished plantations, and farmers were forced into bankruptcy. The worldwide search for gourmet cocoa has rekindled interest in this production, whose fermentation process is a key step in obtaining fine cocoa, thanks to the fact that many processing properties and sensory attributes are developed in this phase. This article presents a blockchain-IoT-based system for the control and monitoring of these events, aiming to catalog in smart contracts valuable information for improvement of the cocoa fermentation process, and future research. Blockchain is used as a distributed database that implements an application-level security layer. A proof of concept was modeled and the performance of the emulated system was evaluated in the OMNet simulator, where a technique based on the SNMP protocol was applied to reduce the amount of data exchanged and resources served/consumed in this representation. Then, a physical platform was developed and preliminary experiments were performed on a real farm, as a way to verify the improvement of the cocoa fermentation process when using a technological approach.

## 1. Introduction

Some important phases are necessary in the production of gourmet chocolate, from the cocoa bean cultivation to the chocolate bar commercialization in the market, and this process depends on several factors such as cocoa farms’ social reality, sustainability goals, natural conditions, type of cocoa variety, expected quality of the final product, etc. These phases can be summarized as follows, where each one of them can be (semi-)automated by computational systems [1]:I.Cultivation—(a) Environmental parameters verification (climate, soil, luminosity, humidity, etc.), (b) land Preparation, (c) seed Selection, (d) seed sowing, (e) irrigation, (f) crop, (g) fertilizing.II.Harvesting—(a) Checking for ripeness, (b) picking, (c) pod and bean separation.III.Processing—(a) Fermentation, (b) drying, (c.1) aging or (c.2) roasting or (c.3) refining and conching, (d) storing.IV.Distribution (supply-chain management)—(a) Transportation, (b) tracking, (c) product quality verification during distribution, (d) warehousing.

As can be observed, there is a lot of work in cocoa processing from its cultivation to its distribution. The improvement in the quality of this process, in such a way to achieve a high-quality cocoa type (fine or gourmet) with well-defined organoleptic properties such as flavor, color, and aroma, is mainly developed in its fermentation and drying processes (items III.a and III.b).

Considering such steps, some articles based on training/calibration approaches have applied colorimetric/imagery techniques to track the state of beans, such as the following: a combination between a colorimetric electronic nose (e-nose) and a near-infrared (NIR) spectroscopy technique to classify cocoa beans in the fermentation step (III.a) [2]; the use of gas chromatography–mass spectrometry (GC–MS) after the application of colorimetry and fluorescence/NIR spectroscopy techniques to measure the progress of fermentation (III.a) and drying (III.b) steps [3]; a platform using an artificial neural network (ANN) approach to investigate the variation of beans colors from images, in order to assess the cocoa fermentation index (FI) (this index is an indirect measure of anthocyanin, which is an indicative about the degree of cocoa bean fermentation) in the fermentation process (III.a) [4]; a proposal using a hyperspectral analysis from chemical changes occurred (pH variation) in the cocoa fermentation process to determine its quality (III.a) [5]; an approach which determines the quality of cocoa beans through their digital images when processed by ANNs (III.c.1) [6].

Other articles monitor certain volatile or liquid compounds produced during some chocolate process phases in order to evaluate its quality, such as in ANNs trained to identify the information required after odors have been trapped by e-noses (composed by arrays of gas sensors used to detect hydrogen, ammonia, ethanol, methane, propane and butane)—(III.c.2) [7] and (III.c.3) [8]; in an application of a GC–MS technique applied after aromatics have been captured from a steam distillation extraction process (III.c.2) [9,10]; in a platform where a NIR spectroscopy technique verifies the cocoa FI through the evaluation of temperatures and amount of ammonia produced in the fermentation step (III.a) [11]; a combination between an electronic tongue (e-tongue, composed of an array of liquid sensors) and multivariate techniques, in order to determine the FI and the titratable acidity (TA) (this property indicates the acid’s impact on the food’s flavor) in grounded cocoa beans, for quality control purposes (III.a) [12].

However, while these proposals are interesting, achieving promising results, they are expensive, time-consuming, and generally involve laboratories which demand extensive sample preparation for calibration processes. In this way, this research aims to offer to cocoa farmers a monitoring real-time solution at a low cost, with the possibility to be manageable remotely, and solutions currently supporting such objectives point to architectures using disruptive technologies such as as Internet of Things (IoT) and blockchain, such as the platforms discussed in the following paragraphs, firstly considering IoT solutions and then blockchain-based distributed proposals.

Reference [13] uses sensory devices to capture temperature and humidity, among other information resulting from cocoa fermentation and drying processes (III.a, III.b), storing them locally in data logger systems for future analysis. Reference [14] proposes an architecture to automatically stir the cocoa mass during the fermentation process in order to improve the quality of beans (III.a), through the use of a microcontroller responsible for orchestrating operations between a temperature sensor, a real-time clock (storing data and the stirring schedule), an AC motor (used for stirring the mass according to the schedule, and whose speed is measured by a rotary encoder sensor), and an LCD display (presenting the previous scheduling and sensor readings—mass temperature and motor rotation speed).

Reference [15] created a solution to aid farmers to supervise several climatic and soil parameters present in cocoa crops, as such luminosity, soil moisture, environment humidity, temperature, etc. (I.a). The information collected by a wireless sensor network is stored, processed, and presented in the web in the format of tables, interactive maps, and statistical graphics, allowing the farmer to be notified in real time if important events take place. Reference [16] developed a system for monitoring the cocoa fermentation process (III.a) through some sensors (of temperature, oxygen, and carbon dioxide) connected to a microcontroller responsible for sending the information captured to the Internet. These data are stored in a database and presented in specialized graphics on a configurable web platform, which allows the adjustment of sensors and the number of wooden/steel boxes to be used in the experiment.

Reference [17] proposes a platform where farmers and researchers monitor climate conditions in order to improve the cocoa production (I.a). Its architecture is composed of cocoa trees equipped with NFC (*near field communication*) tags in such a way that some data can be collected manually; sensor nodes, responsible for capturing environmental parameters (e.g., temperature, humidity, and soil water potential), and a station, responsible for amplifying the signal received from nodes, monitoring the information collected, and storing and transmitting it to the cloud.

Table 1 shows a summary of the characteristics present in such articles (using the IoT concept), comparing them with the architecture proposed in this research, named IoTCocoa (to be discussed in detail in Section 2.2). The following inquiries were investigated in this table: (1) “Uses middleware”, to verify if the platform has a middleware to ease the data management/communication between its software modules and IoT devices. (2) “Allows scheduling the process”, to verify if the platform allows its users to create cocoa beans processing agendas. (3) “Alerts in real time, if necessary”, to verify if the platform is able to trigger alerts in real time, if certain conditions are met. (4) “Allows to insert new process devices”, to verify if the platform allows its users to add new physical devices in its processes (sensors, actuators, fermentation boxes, etc.). (5) “Allows data storage”, to verify if the platform has data storage mechanisms (local and/or web/cloud). (6) “Production phase(s)”, to identify the cocoa production phase(s) addressed by the platform (cultivation, harvesting, processing, distribution). (7) “Interacts with the process (monitoring/intervening)”, to verify if the platform allows its users to monitor (from sensors) and/or intervene (from actuators) some (or several) phase(s) of the chocolate production. As we can observe in this analysis, IoTCocoa is able to answer all features examined, with the exception of characteristic (4).

Considering the use of blockchain and smart contracts technologies in the cocoa production ecosystem, several initiatives have been proposed in the last years in such a way to improve its supply chain transparency (in order to assure the cocoa displacement) and traceability (in order to assure its quality and safety). Such approach promises a better cocoa data manipulation (gathering, storing, and sharing) from the farm to the final client, allowing producers, consumers, and stakeholders to securely access, anywhere, anytime, and from any computational device, the chocolate bar story [18,19,20,21].

Some authors denote the use of such technologies to address unethical activities (e.g., low-income workers, distrust among stakeholders), social violations (e.g., illiteracy, forced labor, gender inequality, poor workplace safety, child trafficking), and agricultural sustainability challenges (e.g., reforestation, overuse of pesticides to the detriment of using natural fertilizers, deforestation caused by illegal cultivation in protected areas) [22,23,24].

Among these proposals, we could cite the Tony’s Chocolonely’s Dutch platform, responsible for sharing cocoa batches information from farms to cooperatives, and onward to exporters, in Ivory Coast, targeting the eradication of child labor and modern slavery in this country [25]. In Colombia, blockchain creates opportunities for investors to directly finance farmers [26]. In Ghana, the exchange of information among cocoa-industry stakeholders assists partnerships to find a better price for this commodity in the local market, with farmers being able to receive their payments directly in their digital wallets [27].

In Ecuador, farmers receive better bargain prices for their harvests (as in Ghana) and rewards (given by European chocolate traders) if the cocoa grains have been manipulated from the use of ethical and sustainable practices. Another initiative based on chocolate produced from Ecuadorian cocoa allows final consumers to have the choice of scanning a powered-token chocolate bar wrapper that can be afterwards used to receive a discount on their next order or to replant a tree in this country [28]. As a last example of the use of this technology, we can mention the Digital Cacau (CAU) cryptocurrency, which follows the commodity channel index of the New York Stock Exchange, where each CAU token is equivalent to 1 kg of cocoa beans [29].

Table 2 shows a summary of the characteristics present in these works (using the blockchain technology), comparing them with the IoTCocoa platform. The following inquiries were investigated in this table: (1) “Uses IoT devices”, to verify if the platform uses IoT devices, in conjunction with the blockchain technology, to interact (monitor/intervene) with the process. (2) “Uses smart contracts”, to verify if the platform uses smart contracts to automatically and securely execute, control and storage events and actions of some (or several) phase(s) of the chocolate production, according to terms established in an electronic agreement. (3) “Is a distributed application (DApp)”, to verify if the platform can be considered as a distributed application (according to [30], a computational environment is considered distributed when all (or part) of its modules are hosted in a decentralized technological infrastructure). (4) “Production phase(s)”, to identify the cocoa production phase(s) approached by the platform (cultivation, harvesting, processing, distribution). (5) “Issue(s) addressed”, to verify the issues addressed by the platform: ethical, social, and/or agricultural sustainability violations, finance (with cryptocurrencies), or, as in the case developed by our proposal, the improvement of the cocoa fermentation process. As we can observe in this analysis, IoTCocoa is able to answer all features examined.

According to the United Nations Conference on Trade and Development (UNCTAD), the chocolate supply chain can be subdivided into five major areas: production, sourcing and marketing, processing of cocoa derivatives, manufacturing and distribution, and retailing to final clients [31]. As we could observe in the last paragraphs, the distributed blockchain applications presented do not address solutions to the cocoa supply chain “production” phase (including the cultivation, harvesting, fermentation, and drying steps), there being, therefore, many opportunities that can still be explored in this domain.

This research contributes to the cocoa fermentation stage. This step is essential for the production of high-quality chocolate. To achieve this goal, we use IoT, blockchain, and smart contracts, considering traditional agricultural strategies.

IoT sensors monitor and collect data in the cocoa fermentation stage. The user creates a schedule in the application where the settings and times to trigger the sensors are defined. When a schedule is started, the application sends requests to the middleware, and the middleware triggers sensors and actuators to collect humidity and temperature values, or trigger actuators, such as driving a motor to stir cocoa beans. IoT devices perform these tasks and send the information back to the middleware.

The middleware sends this information to the smart contracts, in a data manipulation process with properties of non-intermediation, non-repudiation, auditability, immutability, availability, and a certain level of anonymity.

In this way, smart contracts help the construction of IoT-based automation and control systems, eliminating the trusted third party. These contracts store the information on the blockchain, which, in this case, assumes the role of a secure and distributed database. In addition to these contributions, we highlight a modification in the REST architectural style that allows to significantly reduce the exchange of messages between IoT devices and the application.

The rest of this article is organized as follows. In Section 2, we present how farmers, using traditional techniques, manipulate cocoa beans in the fermentation phase in order to produce a chocolate with high quality. After, we propose a platform targeting the optimization of such process through the use of Internet of Things, blockchain, and smart contracts technologies. In Section 3, we evaluate our proposition in two different manners. First, we simulate it through the use of the REST architectural style modified to emulate the SNMP protocol, in such a way to improve the access to network resources, reducing its data traffic. Afterwards, we perform some experiments in a real farm through the use of manual and automated techniques, as a way to verify the improvement of the cocoa fermentation process when using a technological approach (our solution). In Section 4, we situate our proposition in the light of the existing contributions in the fermentation process in the chocolate industry, indicating the advances offered by our work that was emulated, implemented, and tested in a real cocoa farm. We conclude the manuscript presenting some wishes for improvement, to be probably developed and integrated in our solution in future works.

## 2. Materials and Methods

This section details how the process for obtaining gourmet cocoa is performed on farms using traditional/manual techniques (passed down from generation to generation of cocoa farmers); then, we present a new method based on Internet of Things, blockchain and smart contracts, highlighting the role of each of these technologies in the presented proposal.

### 2.1. Traditional Processes in Farms

There is no consensus or standard procedure employed in the (conventional or gourmet) cocoa production around the world. Each farm performs its particular method, sometimes in an empirical way. In southern Bahia (Brazil), for example, some farmers continue to apply the same methods used since the first cocoa plantations took place in this region. Others, however, are trying to experiment with new methodologies developed by regional universities and research centers, such as CEPLAC (Comissao Executiva do Plano da Lavoura Cacaueira—http://www.ceplac.gov.br, (accessed on 7 July 2021)), which has been developing notes and booklets with basic guidelines for conducting cocoa production processes.

In general, cocoa fruits are harvested and broken to remove the pods still in the field; the wet beans are deposited in a container and then transported to the fermentation house, composed of several (wooden or metal) boxes used to start the fermentation process from its lactic, alcoholic, and acetic phases. A fermentation container can be simple, such as in the rectangular sweatbox showed in Figure 1, or more sophisticated, as per the proposals discussed in [32]. Some fermentation containers are cylindrical and can be rotated on the vertical axis, used to mix the cocoa mass.

The wet cocoa beans are initially placed in the slot n.1 for the first 48 h of the process, where the temperature reaches and stabilizes at around 32 °C. After that, the temperature begins to decrease and the beans are removed to the slot n.2, in order to oxygenate the mass and to feed the bacteria responsible for the fermentation process. In this phase, the temperature will increase again.

After a few hours, the oxygen is scarce and the temperature begins to decrease. At this moment, the mass is removed once again to the slot n.1. This mechanism should be repeated until the oxygenation process no longer increases the cocoa mass temperature (step performed between 5 and 7 days), and we could start the drying phase, where the beans are placed in pallets (or patios) in such a way to reduce their levels of acidity and astringency, and to reduce their levels of humidity from 60% to 6–7%, for shipment [33].

In general, temperatures measured in the fermentation process are manually checked twice a day in farms using traditional techniques, with an interval of 10 h between such measures (for example, at 6:00 a.m. and 4:00 p.m), a fact that can represent a hard constraint to the production of a chocolate with high quality. In this way, this research hopes to advance this domain of application from the use of disruptive technologies such as IoT, blockchain, and smart contracts.

### 2.2. IoTCocoa—A New Way to Boost the Gourmet Cocoa Fermentation Process

The IoTCocoa platform arises as a contribution in this domain since it can address such issues in a satisfactory manner [1]. It proposes a continuous monitoring of cocoa beans in the fermentation step, intervening when and if necessary through schedule operations, such as activities duration, setting of fermentation boxes rotational speed, capturing of temperature, humidity, and pH data during such processes, etc., in order to ensure the ideal conditions to the development of organoleptic properties necessary to the production of a chocolate classified as fine, or gourmet. IoTCocoa is able to securely store the information captured in a blockchain, eliminating the trusted third party and allowing auditable, immutable, irrefutable, and available data transitions with a certain level of anonymity.

Its distributed architecture is composed of four modules (see Figure 2, from left to right): Hardware, Middleware, Blockchain, and Application (In this work, the “hardware”, “middleware”, “blockchain”, and “application” words will be written in uppercase when representing IoTCocoa modules (and in lowercase when having a general meaning)). As mentioned previously, a technological platform is considered distributed when all (or part) of its modules are hosted in a decentralized computational architecture [30]. In IoTCocoa, part of its modules are hosted on a traditional web services infrastructure (its front- (Application) and back-end (Middleware) layers), and another part, on a distributed environment (with smart contracts stored in the Blockchain module).

The Application module sends request messages, defined in the process schedule, to the Middleware (step (1)). Next, the Middleware manipulates such information and forwards it to the Hardware module, composed of sensors and actuators that will perform the schedule tasks and send back the result of such operations to the Middleware (step (2)). Afterwards, the Middleware sends such information to smart contracts stored in the Blockchain (step (3)). Finally, Application users will be able to visualize the generated reports and to access the Blockchain to manipulate the data stored on its smart contracts (step (4)).

#### 2.2.1. The Hardware Module

This module constitutes the IoTCocoa hardware layer present in fermentation boxes composed of sensors and actuators compatible with the Arduino platform (see Table 3), as follows: two sensors for measuring cocoa mass temperature (−55 °C/+125 °C), one in the bottom and another in the middle of the box; ambient temperature (−40 °C/+80 °C) and humidity (0–100% UR) sensors; one bean mass pH sensor (0–14); one motor with adjustable speed, for rotating the box; and two relays, for controlling heating/cooling sources (2400 W). These boxes can be used in cocoa farms or labs (in such a way to allow researchers to perform experiments in their universities or research centers).

Two challenges were identified during this stage of development. The first one was the management of processor resources of each sensor/actuator present in the Hardware module, when required simultaneously by different computational calls (e.g., when the cocoa mass temperature sensor is required at the same time for an instruction in the process schedule, and by a user, interacting with the Application web interface).

The second challenge was the implementation of a smart mechanism to avoid sending data captured from sensors if such information has not changed in time. During some experiments, we could observe that such values had small variations in time, and considering this reality, we created a routine responsible for comparing values just collected with data previously captured. If they are equal, the current value is discarded; if different, the captured value is taken into account and sent to the Middleware.

In order to solve these issues, a (2.4 KB) micro-operating system based on threads and priorities was created to manage the execution of such tasks in ESP8266 microcontrollers (ESP8266: a Wi-Fi microchip widely used in IoT projects—https://en.wikipedia.org/wiki/ESP8266, (accessed on 7 July 2021)). Such a micro-operating system is composed of two libraries and three core functions: *threadCocoa()*: to create threads; *setValueTime()*: to adjust the processor usage time, and; *runner()*: to execute threads.

#### 2.2.2. The Middleware Module

The Middleware is the module responsible for receiving processes planning (or agendas/schedules) produced by Application users, and to manipulate and forward its commands to sensors and actuators present in the Hardware layer. Considering the fermentation phase in the cocoa processing approached in this work, the main activities performed by the Middleware are: bean mass temperature and pH readings; ambient temperature and humidity readings; fermentation box rotation; and bean mass cooling/heating. To this end, sensors and actuators should be virtualized in the Middleware, a process which can be operationalized in the following three steps: (a) creation of a sensor/actuator class (with attributes); (b) creation of a class with REST interface mapping; and (c) creation of a data store/read class.

In (a), sensors and actuators are electronically represented by instantiable classes. Each of these classes has variables representing the values to be monitored in the process. In (b), REST interfaces are created based on SNMP messages (to be discussed in the next paragraph), in such a way to effectively manage the network traffic from the application service layer (alternative adopted in [34,35]). Classes in (c) perform database read/write tasks taking into account the information sent by Hardware sensors, or coming from the Application.

The Middleware communicates with the other IoTCocoa modules from JSON messages, in a REST-based architectural style. It has the ability to emulate the SNMP (*simple network management protocol*) protocol (see steps 1 and 2 in Figure 2) [36], and to write, compile, and implement smart contracts in the Blockchain, through data recording operations from information captured by sensors or sent to actuators (see steps 3 and 4 in Figure 2).

The REST style uses the *request*/*response* message exchange pattern, requiring a greater bandwidth and memory/processing capacity, which can directly affect sensors/actuators battery life. However, the proposal discussed in the present article has no such restrictions, since *requests* can be originated from hardware devices or client-side applications, and in either of them, energy restriction is not a critical factor.

The reasons for choosing the REST architecture in detriment of other proposals used in IoT solutions, such as those using the MQTT (message queue telemetry transport) or CoAP protocols, are justified mainly by the following aspects [36]:The actions are performed with HTTP request/response methods for the communication of messages in JSON or XML formats;Applicable in IoT projects thanks to its ease of implementation and interaction with the web and support for M2M commercial platforms in the cloud;Easily implemented on tablets and smartphones for requesting only the HTTP library, available on all operating system distributions and fully utilized in the REST architecture, including cashing, authentication, and content-type negotiation.

In a traditional message mapping using REST, a specific URL is associated with each resource demanded in the network. Hypothetically considering a situation where a schedule requests ambient temperature/humidity and bean mass temperature/pH measures, a REST-based system should map four request messages and wait to receive their respective four responses. In our proposal, the Middleware module maps REST interfaces analogously as done in the SNMP protocol, allowing to retrieve all data requested in this example in a single block, through the use of four SNMP messages: GET (for requesting resources); RESPONSE (for sending resources requested by GET); SET (for sending data to set hardware devices operations); and TRAP (for sending resources not previously requested by a GET). Figure 3 illustrates an example of SNMP messages exchanged between the Middleware and Hardware modules.

In this figure, we can see the Middleware sending a GET message to the Hardware to request some sensors’ values (e.g., ambient temperature, bean mass temperature and pH), and the RESPONSE message sent back from the Hardware to the Middleware, with the requested data (this information will be, afterwards, sent to the Blockchain, to be securely stored on its smart contract(s)). In addition, we can observe, in Figure 3, the special TRAP message (highlighted in orange) that occurs when Hardware elements detect a data modification and automatically send such information to the Middleware, without the need to receive a GET message to perform such task.

The format and fields of this message are detailed in Figure 4a. As we can observe in this representation, this message has fixed and variable length fields. In the fixed header, “IDM” and “IDS” (with 4 bytes each) represent, respectively, the “Middleware Identifier” and the “Sensor Identifier”. “IDT” (with 16 bytes) represents the “Transaction Identifier”, being composed of one incremental field and two registers (R1 and R2). The first transmission is sent with IDT = 0 and R2 = 0. At this moment, R1 stores the size of the variable portion to be forwarded in the next step, when R2 = R2 + R1. In the next transmission, the IDT field will receive the value of R2, and so on. The “Type” field informs the kind of message—Read (L) or Read/Write (G). Finally, “Timestamp” stores the date and time information, and “NVar” indicates, in binary, how many (IDV, Value) pairs compose the message variable field.

The variable header is composed of a “Community” field and a sequence of pairs composed by an “IDV” and a “Value”. The first represents a password (with a maximum length of 64 bytes) that enables read/write operations on Hardware devices, and when combined with the “Type” field, can work similarly to the SNMP protocol. The pairs represent the sensor data. The maximum number of pairs is 255, limited by the “NVar” field.

Figure 4b represents an example of a JSON message sent by Hardware temperature and pH sensors to the Middleware. This format could also be used by external applications with the need for interacting with IoTCocoa, in such a way to establish a common and interchangeable pattern to allow this kind of integration.

#### 2.2.3. The Application Module

The (web service) Application module is composed of two blocks (see Figure 2): *“Schedule Execution Services”*, where users can create and monitor fermentation process schedules from a graphical interface (such agendas will be monitored by smart contracts automatically created by the system); and *“Schedule Result Services”*, where users can generate and visualize reports and dashboards from fermentation process data, retrieved from such smart contracts. The resulting information of this process allows us to verify if the beans produced after this phase will be considered as being of high quality, verified from their aroma, color, and flavor.

A process agenda/schedule describes how Hardware components (sensors and actuators) should behave during the cocoa bean fermentation phase. To this end, the user should define some configuration parameters, such as the following: type of cocoa; times for starting/finishing the process; times for turning on/off mass mixer motors (in order to oxygenate the cocoa mass); times for turning on/off cooling/heating sources; minimum/maximum temperatures for cooling/heating cocoa beans, etc. In addition, he/she should notify about the information of the process to be monitored, such as, for instance, ambient temperature and humidity, cocoa mass temperature, humidity, pH, etc.

The choice about such parameters depends on the country culture. In fact, there is no consensus among gourmet cocoa planters regarding the production of beans with quality. In order to develop a platform answering to farmers’ requirements, some individual interviews and brainstorm sessions were organized, and nine functionalities of the IoTCocoa system were identified and implemented in this work.

#### 2.2.4. The Blockchain Module

Introduced in 2008, the first blockchain network allowed to transact digital values through a distributed structure, introducing the foundations of an alternative economic system based on a digital currency (Bitcoin) [37]. In 2009, the first version of its software architecture provided the elimination of the third-party trust necessary in traditional financial transactions. This technology is scaffolded by three main concepts: encryption, distributed consensus, and ledger, that are ingeniously combined to ensure computational environments able to support the development of cryptocurrencies and decentralized applications.

Encryption satisfies security requirements of applications through the use of cryptographic summaries (or hash functions) and digital signatures. Distributed consensus allows decentralized participants to coordinate their actions and make decisions in order to guarantee the maintenance and progress of the system, despite the existence of probable failures [38]. Ledger is an immutable data structure in which transactions are recorded and the global state of the system is kept replicated across all network distributed nodes.

The blockchain technology guarantees some properties for the development of decentralized applications, such as [38]:Integrity and availability: Data and transactions are replicated to all blockchain participants, keeping the system secure and consistent;Transparency and auditability: Public blockchains are available for anyone to be audited and verified;Immutability and non-repudiation: Blockchain transaction records are immutable (if someone wants to fix them, it will be necessary to create new registers). The use of cryptographic features guarantees the non-repudiation of records;Privacy and anonymity: Transactions are anonymous, based on users’ addresses. Servers store only encrypted fragments of user’s data;Disintermediation: A blockchain eliminates third parties on its transactions, acting as a connector of systems in a reliable and secure way;Cooperation and incentives: A blockchain offers an incentive-based business model in the light of game theory. On-demand consensus is now offered as a service at different levels and scopes;Decentralization: Blockchain applications do not need a central entity to coordinate actions, since their tasks are performed in a distributed manner.

These blockchain characteristics can be used to mitigate several problems existing in modern applications, such as the ones present in IoT systems in Industry 4.0, where we challenge issues of data leakage, security, stability, and reliability [39].

However, before customizing a blockchain layer in an application of this domain, we need to take into consideration some restrictions present in IoT projects, since its devices have several limitations regarding memory space, power consumption, communication rate, and processing performance. In recent years, several works have been published to make this alliance possible [40,41,42].

The reference [39] classifies Blockchain-IoT applications into three groups: (a) digital payment, (b) smart contracts service, and, (c) storage. In (a), the category of digital payments was the first and most widely used in the blockchain field. Currently, monetary blockchains (such as Bitcoin, for instance) can be used on smartphones to process and store part of the data handled in such networks, helping to popularize the manipulation of cryptocurrencies on mobile devices.

In (b), smart contracts are used to build IoT-based automation and control systems, eliminating the trusted third party [43]. Many enterprises provide this kind of service, such as LeewayHertz, which provides solutions for IoT companies and startups interested in operating smart contracts on the Ethereum blockchain [44], and Ecotrace, which uses IoT, artificial intelligence, and the Hyperledger blockchain for tracking food supply chain [45].

Finally, in (c), data warehousing applications see the blockchain technology as a distributed secure database. Factom is one of such examples, that uses a well-defined API (*application programming interface*) to persist information into a blockchain from traditional web platforms [46].

The distributed application (DApp) discussed in this work proposes a Blockchain-IoT solution (as proof of concept), responsible for monitoring gourmet cocoa fermentation processes. Considering the classification presented by [39], our DApp solution offers services in groups (b) (smart contracts) and (c) (storage), both implemented in the Ethereum blockchain.

#### 2.2.5. Smart Contracts

The Ethereum platform allows customers and smart contracts to be represented by unique addresses in the blockchain. A customer account has cryptographic elements responsible for guaranteeing its security, such as password, mnemonic words, and a pair of public and private keys. A smart contract, in turn, has only a representative address, pointing to a location where a computer program is stored. This program cannot execute any action, needing to be invoked by an external agent to perform some of its functionalities, or to trigger other contracts.

Smart contracts need to be compiled and implemented. The compilation process produces two files: bytecodes and ABI (*access binary interface*). Bytecodes represent the compiled contract code that will be sent to a blockchain node. The ABI is a computational interface used by applications to access published contracts, containing instructions for activating their programmed functions, and for manipulating (reading/writing) their properties.

Transactions implementing smart contracts and data manipulation on a blockchain, which include writing or modifying data, come at a financial cost. When a user requests such operations, he/she needs to have an electronic wallet with a sufficient balance (in cryptocurrencies) to carry them out (data reading operations, however, are free of charge).

In this research, we analyzed several software engineering approaches to automate the payment of smart contracts implementation (and manipulation), for which expenses are expected to be paid by the users of the system. Initially, we used the Multiton design pattern, leaving the Middleware responsible for compiling and deploying smart contracts on the Blockchain. This technique guarantees the integrity of the contracts, but presents a drawback: The blockchain network fees will be charged to the Middleware electronic wallet, which should transfer such costs to the respective customers through financial routines to be additionally developed.

A possible approach to mitigate this issue could be carried out through the use of another design pattern (Fabric), where the Middleware continues to be responsible for writing smart contracts each time customers execute schedules in the Application, but now with an additional functionality of sending to the client’s web browser two artifacts: a copy of the smart contract source code and a script, responsible for compiling and implementing such contract on the Blockchain. At the end of this step, the Middleware should receive the contract address and its respective ABI programming interface. The positive aspect of this proposal is the fact that the payment of contract implementation fees is charged directly in each customer’s e-wallet, without the need for controls or additional code development. This technique, however, presents a security breach: a client can modify the smart contract (or script) code before its compilation/deploy in the Blockchain.

Another possibility could be leaving the Application to manage all system e-wallets (one e-wallet for each customer), as performed in [29]. This means that the service provider itself should create, control, and hold the password and public/private keys of e-wallets, allocating a part of its working timeframe to perform such tasks. This solution meets the security requirements regarding the creation and implementation of contracts; however, it requires the development of several additional software routines and graphical interfaces.

The fourth technique proposes another use of the Fabric design pattern, with the Middleware manipulating two different contracts: The administrator contract (CA) and the client contract (CC). The Application instantiates a new CC before executing a new schedule. For this, it sends (as parameters) the e-wallet address of the respective client and control values defined in the scheduler programming. The fees to be paid for this operation will be debited from the customer’s e-wallet, and the new contract will be able to record the information sent by Hardware devices. This one was the approach chosen for us in this research.

After finding a suitable method to charge finance expenses in our blockchain architecture, we will thus be following up with a discussion about the way smart contracts should store processes information. In the first technique, (a), we consider the development of a unique smart contract for each cocoa farm, which should store information about all its harvests. In the second one, (b), we propose a smart contract for each harvest to store the cocoa fermentation processing information from all participant farms, regardless of the farm where such harvest took place.

In the (a) proposition (illustrated in Figure 5), each farm is identified by a different hash code in its smart contract, which is responsible for storing information about several harvests through the identification of their respective *cocoa batch type*, *mass weight*, *price*, *buyer(s)*, and *fermentation process data* (pH, ambient and mass temperatures, activation/deactivation of cooling/heating sources, etc.) This approach creates fewer instances of smart contracts on the Blockchain; however, it needs to store and manipulate a lot of information about several harvests in each farm. In this way, each smart contract could contain many crop arrays, with an amount of data that can increase a lot. Consequently, the system can testify an increment in searching and data record processing, implying the manipulation of additional control routines, which renders the execution of transactions more and more financially expensive.

In the (b) proposition (illustrated in Figure 6), each harvest has an associated contract instance, mitigating the issue of stored information growth in the smart contract array. In this way, it is possible to have a greater predictability and leveling of prices to be paid for contracts manipulation. This figure presents the variables used by the distributed application to correlate schedules and harvests. Among them, we can observe: *contractHash*, a unique key used to identify the contract; *Batch*, that identifies the physical area of the farm where the cocoa beans were harvested (this information can be used in a future integration between our proposal and existing tracking systems); *Size*, a variable informing the batch size (in hectares); *Weight*, that records the weight of cocoa beans sample; *Price*, that registers the price to be paid; *Timestamp*, that records the time when a lot was traded; *Owner*, which registers the contract owner address (and consequently, this one of the lot to be traded); *Buyer*, which represents the lot buyer address; and *Sold*, that informs if a certain lot has already been sold. This one was the approach chosen for us in this research.

## 3. Analysis and Results

According to [47], there are three techniques for systems evaluation: analytical modeling, simulation, and measurement. In this research, we use two of them: simulation and measurement. In the former, we evaluate our proposal through the use of the REST architectural style modified to emulate the SNMP protocol, as a way to reduce the network traffic (explained in Section 3.1). In the latter, we performed some experiments in a real farm through the use of traditional/manual and automated techniques, as a way to verify the improvement of the fermentation process from the use of our Blockchain-IoT solution (explained in Section 3.2).

### 3.1. Simulation Approach—A Preliminary System Performance Analysis

In this research, some tools and frameworks were employed to perform the simulation technique: OMNet (v.5.4.1, to simulate the network); HttpTools (v.0.90, to simulate HTTP message traffic between Middleware and Hardware, and Middleware and Application layers); and INET (v.4.0, an OMNeT framework). To this end, scenarios were created with 1, 5, 10, and 15 simultaneous schedules with 3 sensor behavior profiles (light, moderate, and intense—meaning the information exchanged between Hardware sensors and the Middleware), running in a uninterrupted way for 14 days (average time required to complete the cocoa fermentation and drying processes). Table 4 contains the Middleware configuration values (link speed set to 10 Mbits/s, time in seconds, and information size in KBytes). Table 5 presents the values for the three sensor behaviors, where all *request* messages have size of 165 kBytes.

Figure 7a (left image) shows HTTP sessions opened by the Middleware, considering the use of native REST. Twelve simulations were grouped through the same three sensor behavior profiles. In each one of them, the Middleware was managed by 1, 5, 10, and 15 schedules. It can be observed that in the case of a schedule running for 14 uninterrupted days on the light profile, the Middleware operated without high performance requirements, opening less than 300 sessions. In the intense mode, the value did not exceed 500 sessions. As the number of schedules to be serviced increases, open sessions increase, reaching almost 8000 simulation sessions to 15 schedules in the intense mode.

When using REST with SNMP emulation (Figure 8b, right image), IoTCocoa has a significant reduction in HTTP sessions opened by the Middleware. The test scenario manipulated five schedules in the three sensor profiles. It can be observed that the number of HTTP sessions opened is much lower, and this can be justified by the grouping of requests in single messages. As an example, in the moderate case, whose HTTP sessions are reduced from 2650 to 779, it is possible to note a reduction of approximately 70% in the number of HTTP sessions opened using this configuration.

The number of accesses and resources served/consumed by the Middleware in the three sensor behavior profiles using native REST is shown in Figure 8a (left image). Note that in the light profile, for 1, 5, 10, or 15 schedules, IoTCocoa requires less Middleware processing than other profiles for both access operations and resources served/consumed. When managing 15 schedules in the intense profile, the Middleware was accessed almost 400,000 times and served/consumed over 1,700,000 resources, representing an extremely high value when compared to other tests.

Using these five schedule scenarios for the three profiles, we can observe in Figure 8b (right image) that there is a drop in the number of accesses to the Middleware, in all profiles. The smallest difference, in absolute terms, is found in the light mode with native REST, where accesses represent 16,481 versus 7096 using SNMP emulation mode. In intense mode, the absolute decrease is from 112,870 accesses to 38,322. However, the percentage reduction of accesses in light mode reached 43%, against 33.8% in the intense mode. When checking the resources using SNMP emulation, it can be noted that the biggest absolute difference is found in the intense mode, where there are 609,517 accesses with native REST against 206,127 using SNMP emulation, which represents a 66.2% reduction of the Middleware workload.

### 3.2. Measurement Approach—A Preliminary System Experimental Analysis

For this approach, we considered three lots of cocoa beans (Cocoa beans of same type: Forastero from Baixo Amazonas, Pará, with an intense and citrus flavor) for each experimental method during cocoa fermentation full cycles: traditional (or manual, using classical on-field agricultural techniques) and automated (using the IoT concept with IoTCocoa). In the former, a cocoa cultivator was responsible for manually measuring (cocoa beans and environment) temperatures and the ambient humidity twice a day (6:00 a.m. and 4:00 p.m.). In the latter, sensors were installed in the fermentation box to monitor cocoa beans temperature every 30 min. When the system detected the temperature decrease, it interpreted that the beans needed to be revolved, and set off an alert to farm employees through a red light bulb and an electrical horn. In both techniques there was no automatic mixer to stir the cocoa mass, which was instead moved manually from one side of the fermentation box to the another one (see Figure 1). The end of the fermentation process is identified when the cocoa mass is mixed and the temperature cannot rise, remaining stable or decreasing.

The results found in these experiments are shown in Figure 9. In there, the (a) graphic represents the variation of temperature in the three lots analyzed from the use of the manual approach. In the top of this image, we can observe the cocoa mass temperature range (30–50 °C), and in the bottom, the ambient temperature variation (20–30 °C). In turn, the (b) graphic represents the same experiment of temperature variation, but now taking into account the use of the IoT-based approach (with IoTCocoa). As we can observe in this new image, the cocoa mass (top, 27.5–40 °C) and ambient (bottom, 20–27.5 °C) temperature variations were slightly different when considering the first method.

Temperature monitoring in the fermentation process is a determining factor for the quality of the cocoa produced. In the manual technique, two factors can negatively impact on this final result: The mix of the beans to oxygenate the cocoa mass should be performed only when certain temperature levels are achieved, and the low commitment of farm employees to monitor the process. Delays to revolve the cocoa mass can lead to a sub-fermented product with a high amount of violet beans that alter the final chocolate flavor, while also allowing the creation of fungi that can affect the whole lot.

Therefore, farm workers should have a fundamental role in the manual process but, unfortunately, this is not what happens in practice, as can be observed in Figure 9a, where ambient temperatures collected in the “ambient temperature for lot 1 (bottom)” curve should present distinct values in different hours of a day: one in the early morning at 6:00 a.m., theoretically colder, and another at 4 p.m., theoretically warmest. However, it is possible to note that the temperature remains constant during three consecutive captures (30 °C). This may lead one to infer that the measurements were not performed properly, as the ambient temperature is unlikely to remain at the same value for almost 24 h.

Other issues related to monitoring activities, performed by human beings, are as follows: Firstly, the fact that the fermentation is a continuous process taking place on Saturdays, Sundays, holidays, and weekday times between, for instance, 4 p.m. and 6 a.m, periods in which farm employees should work, but that in practice, there are not guarantees if such measures should be properly performed. Second, the inaccuracy of the temperature data collected inside the fermentation container: each measure performed by the farm worker can be made in a different position of the box.

In the IoTCocoa approach, in turn, we could observe a homogeneity in cocoa mass and ambient temperature curves, and this can be explained by the following three reasons: Firstly, measurements performed every 30 min allow the construction of a smoother graph. Second, and this is only valid for the group of curves representing cocoa mass temperatures (curves in the top of (b)), since farm workers can quickly react to revolve the cocoa mass from (light and sound) signals received by the alarm system, beans do not need to wait for hours until they can be reoxygenated and the fermentation process can continue. Third, the accuracy of temperature data collected at the same points of the fermentation container (always in the bottom and the middle of the box).

The IoTCocoa architecture was conceived to eliminate the issues (in-)directly caused by the human presence in the manual fermentation process, automatically collecting sensors data and activating/checking actuators from an agenda of predefined times, in such a way to ensure that the cocoa masses have adequate tools to identify the time to mix the beans and to allow the fermentation process to have the adequate conditions to execute.

We can observe in the three lots of each approach an interesting feature: in the manual approach, the three fermentation experiments had different finish times. Lot 1 was the shortest, followed by Lot 3, and finally Lot 2, whose fermentation period was the longest. With the IoTCocoa process, however, the three lots had the same time range, fact that leads us to believe that we can expect fermentation processes with more standardized results.

Figure 10 presents the relative humidity of air measured during the fermentation of Lots 4, 5, and 6, in the IoTCocoa approach. These measures are innovative since they are not systematically collected during the fermentation process, allowing from now on the realization of new studies to identify the effects caused by very humid environments in such contexts (such as, for instance, the relative humidity of air in the region where such experiments performance can reach 94%). Among these effects, it is worth stressing the fungi production and poor fermentation effects. Fungi can proliferate in cocoa beans if the environment is very humid, developing undesirable odors, tastes, and stains in the beans. Poor fermentation can occur because one of the bacteria responsible for performing this process may not be able to raise the temperature to appropriate levels in very humid environments.

## 4. Discussion and Conclusions

In this research we proposed an automated solution targeting the improvement of fermentation processes in cocoa farms, through the use of IoT and blockchain technologies. As we could observe in Section 1, well-established IoT architectures are already in use nowadays, offering a better result of the cocoa produced using this approach [13,14,15,16,17].

However, the use of the blockchain technology in such domain is still in its embryonic phase [48], even considering its secure application in the chocolate distribution phase, already performed by some enterprises that have been rewarding their farm partners using ethical and sustainable business practices [25,26,27,28,29].

One of the main issues causing this situation is the reduced number of stakeholders controlling the chocolate industry in the world, which generally do not permit the introduction of new practices in their consolidated processes [49]. Consequently, some authors have estimated the full use of this technology in the cocoa supply chain only in the next 5–10 years [20,25].

Other issues are associated with the handling of cocoa beans in the field, mainly caused by the lack of evolution of harvesting and processing (fermentation/drying) methodologies, all of extreme importance to the production of a chocolate with high quality [50]. Taking into consideration such problems, we developed in this work a monitoring Blockchain-IoT architecture at a low cost, named IoTCocoa. This innovative real-time platform allows a remote intervention of its users through schedule operations, when and if necessary, being able to satisfactorily address today, and not only in the coming decade, the issues existing in the gourmet cocoa fermentation phase [1].

Its distributed architecture is composed of four modules (see Figure 2 in Section 2): Hardware, Middleware, Blockchain, and Application. The Application module sends request messages, defined in the process schedule, to the Middleware, which manipulates such information and forwards it to the Hardware module, composed of sensors and actuators that will perform the schedule tasks and send back the result of such operations to the Middleware. This module sends such information to smart contracts (created by the Application and stored in the Blockchain). Finally, Application users will be able to visualize the generated reports and to access the Blockchain to manipulate the data stored in its smart contracts.

Each Hardware device is composed of a micro-operating system conceived to aid the processors of sensors and actuators to better manage its resources, if required simultaneously by different computational calls, and avoid sending data captured if such information has not changed in time (less network traffic, less contract operations to be performed).

The Middleware module communicates with the other IoTCocoa modules from JSON messages, in a REST-based architectural style. It has the ability to emulate the SNMP protocol and to write, compile, and implement smart contracts in the Blockchain through data recording operations from information captured by sensors, or sent to actuators, according to the schedules defined by Application users.

The Application module allows users to create and monitor fermentation process schedules from graphical interfaces, and also generate and visualize reports and dashboards resulting from the execution of such agendas. Such transactions are securely carried out with the aid of smart contracts created by the Application, and the fees to be paid for such operations are directly withdrawn from customers’ e-wallets.

This proposition was evaluated in two different manners. First, we simulate our system using the REST style modified to emulate the SNMP protocol, in such a way to improve the access to network resources, reducing the web data traffic. In this experiment, we were ableto reduce significantly the number of HTTP sessions opened by the Middleware, thanks to the grouping of request demands in single messages (approximately 70% in some network configurations), as well as a decrease in its number of access operations (approximately 66% in some network configurations) and resources served/consumed (approximately 66% in some network configurations, which represents a significant reduction of the Middleware workload).

In the second evaluation, we performed some experiments in a real farm through the use of manual and automated techniques as a way to verify the improvement of the fermentation process when using a technological approach, conceived to eliminate the issues (in-)directly caused by the human presence in traditional methodologies. In there, schedule instructions are run to automatically collect data from sensors and to activate/check actuators in predefined times.

As future work, the authors of this research aim to develop in the next versions of IoTCocoa:Manuals/booklets, to better qualify farmers’ employees to be able to work with its technologies.Cocoa fermentation schedule patterns, conceived from good practices established in several countries around the world (e.g., Ivory Coast, Ghana, Ecuador, Indonesia, Brazil, etc.).The integration of automated mixers in another version of the Hardware module, in such a way to better oxygenate the cocoa mass in the fermentation process.Its improvement in order to support other steps of the chocolate production industry (e.g., drying, roasting, storing, tracking, etc.), as well as its application in other crop cultures (e.g., coffee, wheat, corn, sugarcane, etc.).The integration of artificial intelligent mechanisms (e.g., ontologies, neural networks, etc.), in order to improve the chocolate supply chain.

## Figures and Tables

**Figure 1 sensors-22-03029-f001:**
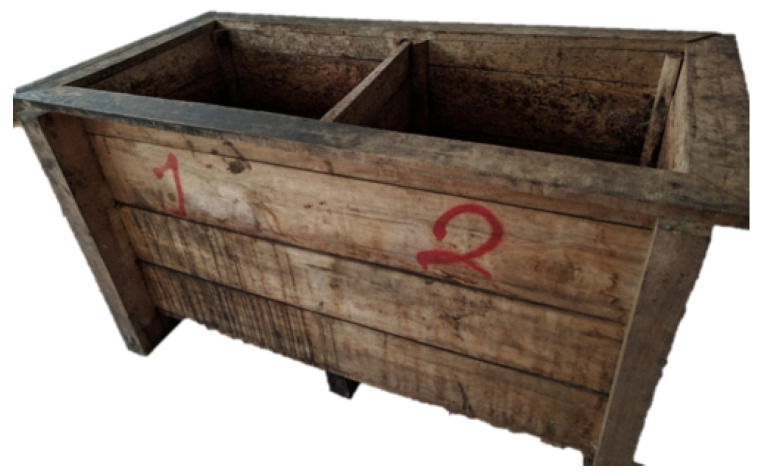
A wooden sweatbox (with 2 slots) used in the cocoa fermentation step.

**Figure 2 sensors-22-03029-f002:**
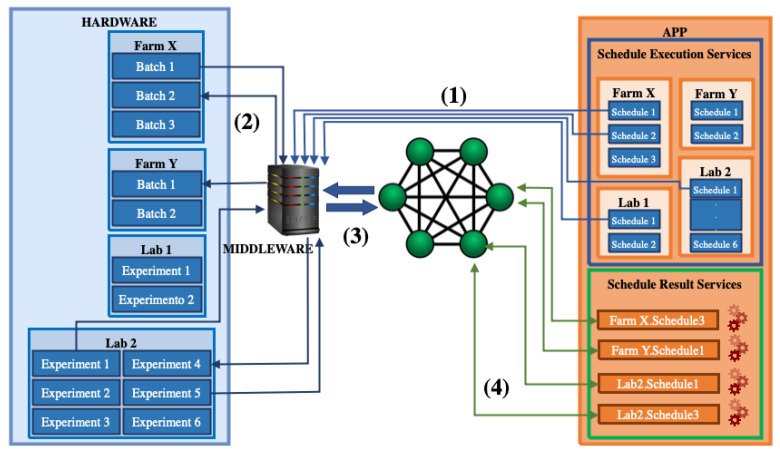
IoTCocoa architecture (module view).

**Figure 3 sensors-22-03029-f003:**
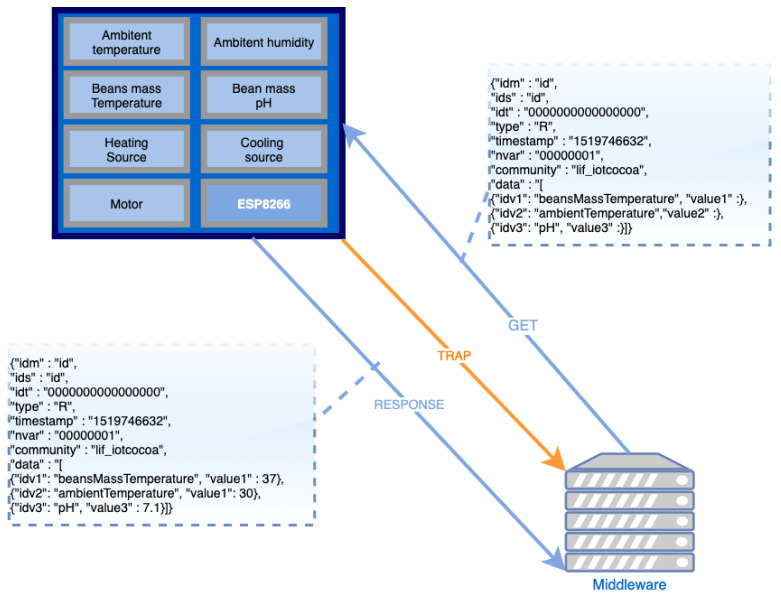
SNMP messages exchanged between the Middleware and Hardware modules.

**Figure 4 sensors-22-03029-f004:**
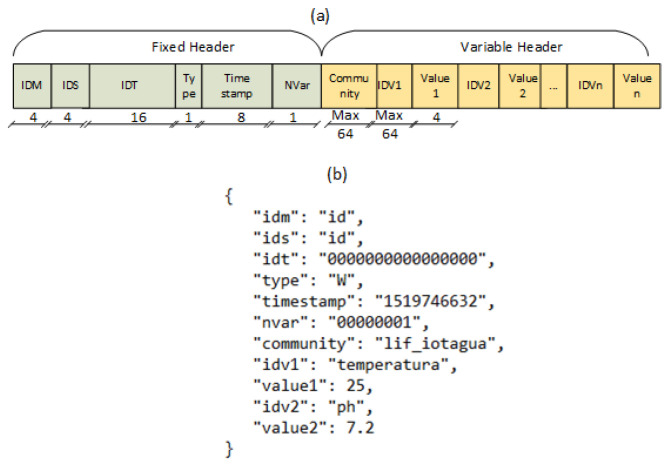
(**a**) Message format; (**b**) example of an IoTCocoa JSON message pattern.

**Figure 5 sensors-22-03029-f005:**
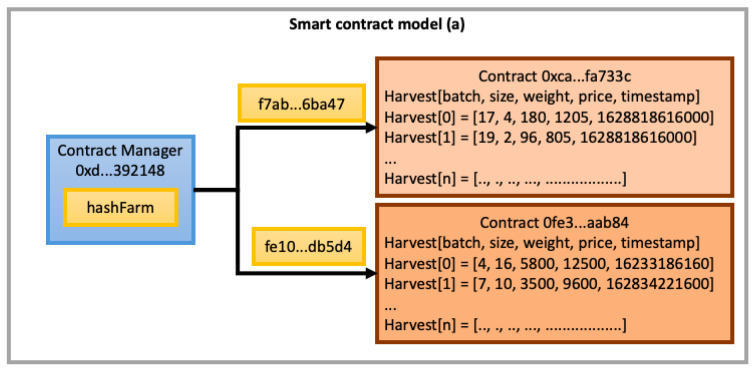
One smart contract for each farm proposition.

**Figure 6 sensors-22-03029-f006:**
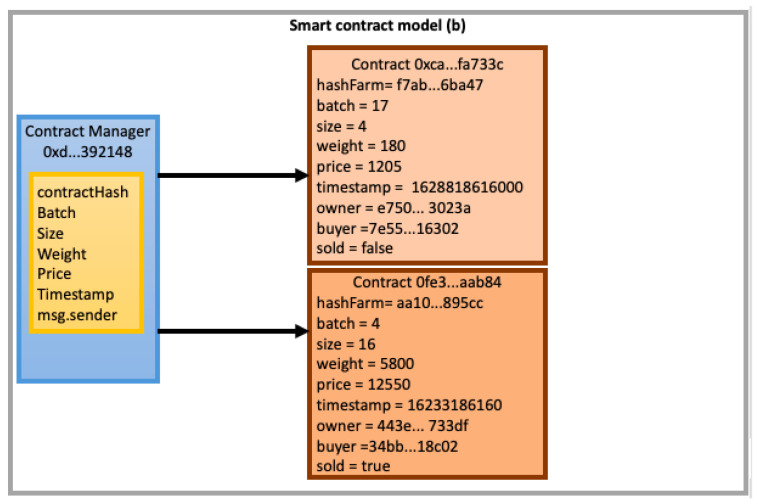
One smart contract for each harvest proposition.

**Figure 7 sensors-22-03029-f007:**
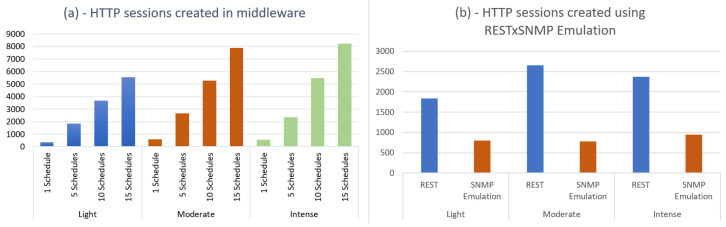
HTTP sessions opened by the Middleware using (**a**) REST architectural style; (**b**) REST with SNMP emulation.

**Figure 8 sensors-22-03029-f008:**
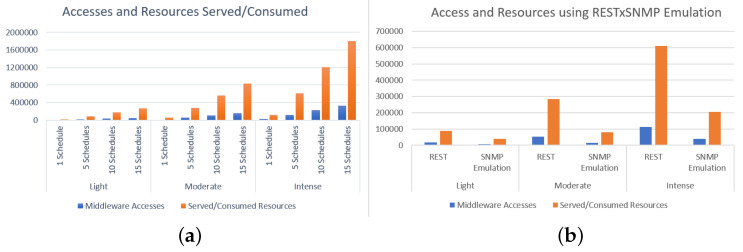
(**a**) Accesses and resources served/consumed by the Middleware. (**b**) HTTP sessions opened using REST x REST with SNMP emulation.

**Figure 9 sensors-22-03029-f009:**
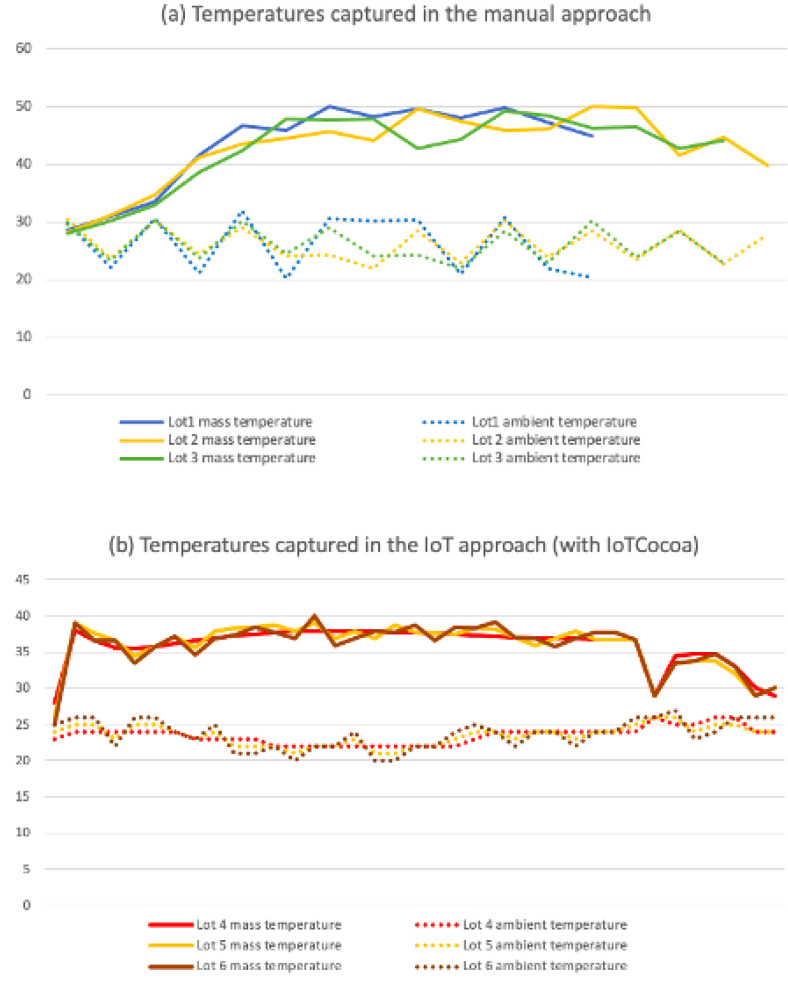
Cocoa mass and ambient temperature variation captured from (**a**) manual and (**b**) automated techniques.

**Figure 10 sensors-22-03029-f010:**
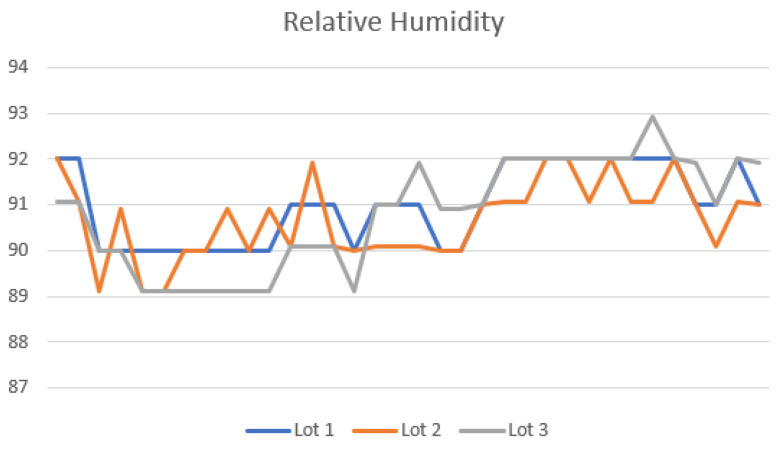
Relative humidity using the IoTCocoa approach.

**Table 1 sensors-22-03029-t001:** Comparison between IoTCocoa and other cocoa IoT-based platforms.

	[13]	[14]	[15]	[16]	[17]	IoTCocoa
(1) Uses middleware						Yes
(2) Allows scheduling the process		Yes				Yes
(3) Alerts in real time, if necessary			Yes			Yes
(4) Allows to insert new process devices				Yes		
(5) Allows data storage	Local	Local	Web	Web	Web	Web
(6) Production phase(s)	Processing (fermentation, drying)	Processing (fermentation)	Cultivation (environmental parameters verification)	Processing (fermentation)	Cultivation (environmental parameters verification)	Processing (fermentation)
(7) Interacts with the process (monitoring/ intervening)	Humidity, mass temperature	Mass temperature, rotation motor	Luminosity, soil, moisture, ambient temperature and humidity	Oxygen, carbon dioxide, mass temperature	Soil (water potential), ambient temperature and humidity	Mass pH and temperature, ambient humidity and temperature, rotation motor, heating and cooling sources

**Table 2 sensors-22-03029-t002:** Comparison between IoTCocoa and other cocoa blockchain-based platforms.

	[25]	[26]	[27]	[28]	[29]	IoTCocoa
(1) Uses IoT devices						Yes
(2) Uses Smart Contracts	Yes				Yes	Yes
(3) Is a Distributed Application (DApp)	Yes	Yes				Yes
(4) Production phase(s)	Distribution (tracking)	Distribution (tracking)	Distribution (tracking)	Distribution (tracking)		Processing (fermentation)
(5) Issue(s) addressed	Ethical and social violations	Ethical and social violations	Agricultural sustainability violations	Agricultural sustainability violations	Finance	Improve the cocoa fermentation processing

**Table 3 sensors-22-03029-t003:** Sensors and actuators technical features (Hardware module).

Sensors/Actuators	Technical Features
Bean mass temperature	−55 °C–+12 °C
Ambient temperature	−40 °C–+80 °C
Ambient humidity	0–100% UR
Bean mass pH	pH between 0–14
Rotation Motor	Adjustable Speed
Heating Source	Power of 2400 W
Cooling Source	Forced Ventilation

**Table 4 sensors-22-03029-t004:** Middleware configuration values.

Values	Distribution Type	Mean	Min.	Max.	SD
Page Size	Fix		3000		
Delay	Normal	0.05	0.01		0.01
Resource size	Exponential	430	165	600	

**Table 5 sensors-22-03029-t005:** Values used in the simulation for all three traffic types.

	Sensor Profile: Light—Moderate—Intense		
	**Mean**	**Min**	**SD**
Activity Period	21,600-43,200-84,600	7200-288,00-84,600	1000-10,800-0
Request Interval	900-600-300	60-60-150	60-60-60
Session Interval	3600-3600-100	120-120-60	900-10,800-20
Number of Requests/Session	5-10-100	2-2-5	1-5-50
Processing Delay	0.05-0.05-0.05		2-0.01-0.01

## Data Availability

Not applicable.

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
