# Peer review of "Improving Data Security with Blockchain and Internet of Things in the Gourmet Cocoa Bean Fermentation Process†"

_sensors, 2022, doi:10.3390/s22083029_

Round 1

Reviewer 1 Report

  1. The thing that is discussed most in this paper is from the application side. Please add more discussion regarding IoT and hardware( type of sensors and brand too) which are needed to be deployed.
  2. How to deal if the connection is lost or bad? data can't be sent to the server/middleware. Also please mention some references, some studies have already been solved these problems.
  3. The explanation for Figure 2 needs to be improved and the image needs to be detailed regarding the IoT.

Reviewer 2 Report

In the abstract, it is not clear how security is achieved using blockchain. Moreover, quantitative analysis is also missing.

All the references inside the text are not visible. what is meant by ? during citing the references?

I suggest to authors rewrite the introduction section and highlights the contributions of their proposed work.

the related work section has also not been seen in the manuscript. It would be better to include it as a separate section. Also, highlight the contributions and limitations of the existing work in the form of a comprehensive table.

I have not seen any block diagram/flow diagram of the proposed work. It must be included.

The results are trivial. The authors should perform some more experiments along with their discussion.

the conclusion section is not seen in the manuscript. It must be included as a separate section before references. 

I have not seen the limitations of the proposed work? It should also be included in the conclusion part. What is the future work for the proposed solution?

Reviewer 3 Report

The authors present a case study about the use of blockchain technology in agriculture. Their goal is to provide a low-cost, real-time monitoring tool, specifically for cocoa.

Please note that all reference numbers are missed (they do not appear in the text). The numbers of Tables, Figures, and references to Section numbers are also missed.

The authors mention "anonymity" in line 246, why is that required (if so) in this scenario?

Are the fermentation boxes mentioned in line 264 the same as shown in the figure? How is the fermentation box that includes the hardware explained in section 2.2.1? What is the communication protocol used to receive the messages/Actions and to send the captured data? how is energy provided to these IoT devices? How long do they live?

It is not clear for me after reading the paper the use of SNMP, please explain deeply. What is the improvement versus a classical web approach? Or use an alternative as MQTT?

There is a clear overlapped between this work and reference [https://doi.org/10.1109/LATINCOM48065.2019.8937903]. (It is very similar and some figures/tables seem repeated.) The only new sections are 2.2.4 and 2.2.5. Then results of section 3.1 are exactly the same between this paper and reference, which could be considered self-plagiarism. Results included in section 3.2 are original, however, there are no results regarding the use of blockchain. The introduction, abstract, and title and misleading then, since no PoC is done in this sense (as far as I see), and unfortunately also decreases the novelty and contribution of this work.

I strongly recommend the authors shorten the paper, summarizing the contents in all sections but 2.2.4, 2.2.5, 3.2, and 4 (and very importantly avoiding any type of self-plagiarism). Then, I recommend redoing the introduction and abstract and clearly stating if blockchain is used or not, to what extent, and if results are provided or not (experiments? simulations?). Finally, those new sections I humbly think that they require a major revision, explaining better their content.
